# Fucoxanthin Ameliorates Sepsis via Modulating Microbiota by Targeting IRF3 Activation

**DOI:** 10.3390/ijms241813803

**Published:** 2023-09-07

**Authors:** Jingqian Su, Biyun Guan, Qiaofen Su, Shan Hu, Shun Wu, Zhiyong Tong, Fen Zhou

**Affiliations:** 1Fujian Key Laboratory of Innate Immune Biology, Biomedical Research Center of South China, College of Life Sciences, Fujian Normal University, Fuzhou 350117, China; gby936552696@126.com (B.G.); 15260328242@163.com (Q.S.); qbx20220152@yjs.fjnu.edu.cn (S.H.); qsx20221412@student.fjnu.edu.cn (S.W.); tongzhiyong1998@163.com (Z.T.); 15859446856@163.com (F.Z.); 2Provincial University Key Laboratory of Microbial Pathogenesis and Interventions, College of Life Sciences, Fujian Normal University, Fuzhou 350117, China

**Keywords:** sepsis, aloe emodin, inflammation, cecal ligation puncture, gut microbiota

## Abstract

To improve patient survival in sepsis, it is necessary to curtail exaggerated inflammatory responses. Fucoxanthin (FX), a carotenoid derived from brown algae, efficiently suppresses pro-inflammatory cytokine expression via IRF3 activation, thereby reducing mortality in a mouse model of sepsis. However, the effects of FX-targeted IRF3 on the bacterial flora (which is disrupted in sepsis) and the mechanisms by which it impacts sepsis development remain unclear. This study aims to elucidate how FX-targeted IRF3 modulates intestinal microbiota compositions, influencing sepsis development. FX significantly reduced the bacterial load in the abdominal cavity of mice with cecal ligation and puncture (CLP)-induced sepsis via IRF3 activation and increased short-chain fatty acids, like acetic and propionic acids, with respect to their intestines. FX also altered the structure of the intestinal flora, notably elevating beneficial Verrucomicrobiota and *Akkermansia* spp. while reducing harmful *Morganella* spp. Investigating the inflammation–flora link, we found positive correlations between the abundances of *Morganella* spp., *Proteus* spp., *Escherichia* spp., and *Klebsiella* spp. and pro-inflammatory cytokines (IL-6, IL-1β, and TNF-α) induced by CLP. These bacteria were negatively correlated with acetic and propionic acid production. FX alters microbial diversity and promotes short-chain fatty acid production in mice with CLP-induced sepsis, reshaping gut homeostasis. These findings support the value of FX for the treatment of sepsis.

## 1. Introduction

Sepsis, characterized by life-threatening organ dysfunction or failure, is linked to an aberration in the equilibrium between inflammatory responses and immune system suppression [1]. The activation of cytokines, complement components, and the coagulation system contributes to the excessive inflammatory response associated with sepsis [2,3,4]. Despite ongoing advancements in the clinical management of sepsis, specific pharmaceutical interventions are lacking. Sepsis-inducing infections are the leading cause of mortality among critically ill surgical patients [5]. Previous clinical studies have shown that a cytokine storm, involving a significant systemic release of proinflammatory cytokines in animal experiments, leads to systemic inflammation [6]. Eliminating proinflammatory cytokines reduces organ damage and mortality associated with inflammation [7].

The cecal ligation and puncture (CLP) model is used extensively because it closely mimics the clinical progression of sepsis in humans [8]. When the CLP model is applied, intestinal bacteria, fungi, and metabolites migrate into the abdominal cavity, leading to abdominal infections and systemic sepsis [9].

The gut microbiome plays a pivotal role in maintaining body homeostasis, exerting a significant influence on pathogen defense, food digestion and absorption, and immune system regulation [10,11]. It exhibits a robust correlation with the onset of sepsis [12,13].

Fucoxanthin (FX), a naturally occurring carotenoid derived from brown algae, is a promising drug candidate for treating various diseases, including sepsis, with minimal toxicity and adverse reactions [14]. Previous studies have demonstrated that FX significantly reduces mortality in a CLP-induced sepsis mouse model via interferon regulatory factor 3 (IRF3), effectively suppresses pro-inflammatory cytokines and ROS, improves pathological damage, and activates autoimmune cells [15,16,17,18].

However, the precise mechanisms underlying the regulatory role of FX-targeted IRF3 in modulating the composition of the bacterial flora and its subsequent impact on sepsis development remain unclear. To examine the impact of FX on peritoneal and intestinal microbes in mice, we generated CLP sepsis models in wild-type (WT) and *Irf3^−/−^* mice. This study aims to establish a novel theoretical framework for utilizing FX in sepsis treatment.

## 2. Results

### 2.1. Effect of FX on Mice with CLP-Induced Sepsis via the Inhibition of IRF3

CLP-induced sepsis was modeled in mice with FX (1.0 mg/kg/day). In WT mice, the group treated with FX (1.0 mg/kg/day) exhibited decreased agglomeration, enhanced activity, and improved appetite than those in the untreated CLP group. Conversely, in *Irf3^−/−^* mice, there were no statistically significant disparities in physical parameters between the CLP + FX and CLP groups (Figure 1a,b).

In a CLP-induced sepsis mouse model, FX suppressed pro-inflammatory cytokine levels by inhibiting IRF3. The lung tissues of mice in the CLP group demonstrated higher expression levels with respect to pro-inflammatory cytokines IL-1β, IL-6, and TNF-β at both the protein and mRNA levels than those in the Control group. After FX treatment, WT mice subjected to CLP showed significantly lower protein and mRNA expression levels with respect to pro-inflammatory cytokines than those in the CLP-treated mice (*p* < 0.0001; Figure 1c–e). Conversely, no significant inhibitory effect was observed in *Irf3^−/−^* mice (*p* > 0.05; Figure 1f–h). These results support the inhibitory effect of FX on inflammation in mice with CLP-induced sepsis. Additionally, the observed effects were IRF3-dependent.

### 2.2. FX Can Effectively Reduce Bacterial Counts in the Abdominal Cavity of Mice with CLP Sepsis via IRF3

To investigate the effect of FX on the abundance of microorganisms within the abdominal cavity of CLP mice, the ascites of WT mice was diluted 10^5^ times and that of *Irf3^−/−^* mice was diluted 10^4^ times based on the dilution ratio determined during the preliminary experiment. The coated samples were then subjected to microbial growth and colony enumeration.

As shown in Figure 2 for WT mice, minimal colony growth was observed in the plates of the Sham and Sham + FX groups. Conversely, a substantial number of colonies were observed in the plates for the CLP group (*p* < 0.0001). Notably, the number of colonies in the plates of the CLP + FX group was significantly lower than that in the CLP group (*p* < 0.0001).

For *Irf3^−/−^* mice, no colony growth was detected in the plates of the Sham and Sham + FX groups (Figure 2). Conversely, a substantial number of colonies were observed in the plates of the CLP group (*p* < 0.0001). However, there was no significant difference in the number of colonies in the plates of the CLP + FX and CLP groups (*p* > 0.05).

These findings indicate that CLP induces a lasting intraperitoneal microbial infection in mice, while FX demonstrates reduces the quantity of intraperitoneal microbial bacteria in mice with CLP sepsis by activating IRF3, thereby achieving an anti-bacterial effect.

### 2.3. FX Can Effectively Reduce the Content of Acetic Acid and Propionic Acid in the Peritoneal Lavage of CLP Sepsis Mice via IRF3

According to the data presented in Figure 2k,l, the absence of acetic acid and propionic acid was observed in both the Sham and Sham + FX groups. In WT mice, the levels of acetic acid and propionic acid in the CLP group were significantly higher than those in the Sham group (*p* < 0.0001). Conversely, acetic acid and propionic acid levels were significantly decreased following FX administration (*p* < 0.0001). In *Irf3*^−/−^ mice, the levels of acetic acid and propionic acid in the CLP group were significantly higher than those in the Sham group (*p* < 0.01). However, the levels of acetic and propionic acid did not change significantly after FX treatment (*p* > 0.05). FX had a strong inhibitory effect on acetic acid and propionic acid contents in the peritoneal lavage of mice with CLP sepsis, and these effects were mediated by IRF3.

### 2.4. FX Regulated Intestinal Flora Homeostasis via IRF3

To investigate the effect of FX on the intestinal flora of WT and *Irf3*^−/−^ mice, we used third-generation 16S RNA sequencing technology to examine variations in microbial communities among the groups. A Venn diagram shows that the number of operational taxonomic units (OTUs) unique to WT mice in the Sham, Sham + FX, CLP, and CLP + FX groups was 1, 10, 6, and 1, respectively, while the number of OTUs unique to *Irf3*^−/−^ mice in the Sham, Sham + FX, CLP, and CLP + FX groups was 4, 2, 5, and 5, respectively (Figure 3a). Analyses of alpha diversity indices, including Chao1, Shannon, Simpson, and Ace, revealed significant differences between the CLP and CLP + FX groups of WT mice and the *Irf3^−/−^* mice (Figure 3b–e). Next, we performed a beta diversity analysis based on weighted UniFrac distances, as illustrated in Figure 3f,g. A PCoA revealed a significant separation of gut microbial communities between the eight groups of WT and *Irf3^−/−^* mice (*p* < 0.05), and ANOSIM confirmed these results (*p* < 0.05).

To further investigate variations in the microbiota structure, genus-level abundances of microbes were analyzed (Figure 4). As shown in Figure 3h–j, *Akkermansia* spp. dominated the Sham group of both WT and *Irf3*^−/−^ mice. Among the most abundant genera in the Sham + FX group of both WT and *Irf3*^−/−^ mice were *Akkermansia*, uncultured Muribaculaceae, *Escherichia*, *Shigella*, *Bacteroides*, *Lactobacillus*, *Clostridium* sensu stricto 1, and Lachnospiraceae NK4A136. In WT mice, *Akkermansia* and *Escherichia* spp. were most abundant in the CLP group. Conversely, the abundance of *Akkermansia* spp. increased significantly after FX treatment in the CLP + FX group (*p* < 0.05). In *Irf3*^−/−^ mice, CLP + FX mice did not display a notably higher abundance of *Akkermansia* spp. than that of *Irf3^−/−^* mice (*p* > 0.05).

### 2.5. Linear Discriminant Analysis Effect Size (LEfSe) of the Intestinal Microbiota

The microbiota in the Sham, CLP, and FX treatment groups were examined via an LEfSe analysis (Figure 5a). Using linear discriminant analysis (LDA) and a score of >4 as the screening condition, 42 species with significant differences in information were detected. *Lactobacillus* was the predominant bacterial genus in the Sham group. *Clostridia*, *Clostridiales*, and *Escherichia* were more abundant in the CLP group than in the Sham group. *Bacteroides* and *Muribaculum* were more abundant in the FX group than in the CLP group.

In total, 42 biomarkers differed significantly from the phylum to the species level among the three groups of samples assessed and were mainly distributed in Bacteroidetes, Firmicutes, Proteobacteria, and Verrucomicrobia (Figure 5b). At the genus level, the leading bacterial genera in the Sham, CLP, and FX groups were *Lactobacillus* and *Akkermansia*; *Escherichia* and *Lachnospiraceae*; and *Bacteroides* and Muribaculaceae. These results indicate that FX alters the dominant intestinal microbial taxa and reshapes the structure of the intestinal microbiota in mice with sepsis.

### 2.6. FX Affects the Function of Intestinal Flora in Mice with CLP Sepsis via IRF3

To investigate the effect of FX on the function of the intestinal flora in mice with CLP-induced sepsis, we used PICRUSt2 to predict functional genes in the CLP and CLP + FX groups and found seven metabolic pathways with significant differences between groups (Figure 6). The top three significant differences were observed with respect to nucleotide metabolism, environmental adaptation, and carbohydrate metabolism. The three most abundant genes were involved in carbohydrate metabolism, nucleotide metabolism, and the cellular community of prokaryotes. Nucleotide metabolism was observed in the CLP group, while carbohydrate metabolism occurred in the CLP + FX group, which may be the main mechanism by which FX affects the intestinal microflora in CLP sepsis.

### 2.7. FX Promoted the CLP-Induced Production of SCFAs in the Intestinal Flora of Mice with Sepsis via IRF3

Short-chain fatty acids (SCFAs) are the major metabolites of the intestinal flora and are strongly associated with inflammatory and immune responses in the host. As shown in Figure 7a,b, in WT mice, the levels of both acetic and propionic acids were remarkably lower in the CLP group than in the Sham group (*p* < 0.0001), whereas the levels of both acids were higher in the CLP + FX group than in the CLP group (*p* < 0.0001). In *Irf3^−/−^* mice, compared with levels in the Sham group, the levels of both acids were remarkably lower in the CLP and CLP + FX groups (*p* < 0·0001). These results revealed that the effect of FX treatment in *Irf3^−/−^* mice differed significantly from that in WT mice.

To further investigate the relationship between the changes in the intestinal flora and pro-inflammatory factors, we searched for significant differences in species composition between WT and *Irf3^−/−^* mice. As shown in Figure 7c,d, the mRNA and protein expression of pro-inflammatory factors were positively correlated with *Morganella* spp. (*p* < 0.001) in WT mice and were positively correlated with *Bacteroides* spp., *Helicobacter* spp., and *Lysinibacillus* spp. in *Irf3^−/−^* mice (*p* < 0.05). *Escherichia* spp. and *Klebsiella* spp. were positively correlated with the protein levels of TNF-β, IL-1β, and IL-6 in WT mice (*p* < 0.001) and with *Clostridium* sensu stricto 1 and *Bacillus* spp. in *Irf3^−/−^* mice (*p* < 0.001). The serum levels of IL-1β, IL-6, and TNF-α were negatively correlated with *Parasutterella* spp. and *Roseburia* spp. in WT mice; and with *Akkermansia* spp., *Ligilactobacillus* spp., *Turicibacter* spp., and uncultured Bacteroidales in *Irf3^−/−^* mice (*p* < 0.01).

In WT mice, *Lachnoclostridium*; Lachnospiraceae NK4A136 group; and *Lactobacillus*, an uncultured bacterium from Muribaculaceae, *Parasutterella*, and *Roseburia* were significantly positively correlated with acetic and propionic acids, whereas *Morganella*, *Proteus*, *Escherichia*, and *Klebsiella* were significantly negatively correlated. In *Irf3^−/−^* mice, *Lactobacillus*, *Parasutterella*, and *Ligilactobacillus* showed significant positive correlations with acetic and propionic acids (*p* < 0.01), whereas *Clostridium* sensu stricto 1, unclassified Lachnospiraceae, and unclassified Oscillospiraceae were significantly negatively correlated (*p* < 0.01). Interestingly, these changes were opposite to the changes in inflammatory factor levels observed in serum and lung tissues.

As shown in Figure 7e, redundancy analysis and canonical correspondence analysis were used to analyze the correlations between SCFAs and bacterial populations at the genus level. Acetic and propionic acids were positively correlated with the Lachnospiraceae NK4A136 group, *Lactobacillus*, and *Akkermansia* and negatively correlated with *Morganella*, *Alloprevotella*, *Proteus*, *Klebsiella*, *Escherichia*, and *Bacteroides*. A higher correlation was observed between *Morganella* and CLP groups, *Alloprevotella* and CLP + FX groups, and Lachnospiraceae NK4A136 and the Sham and Sham + FX groups.

## 3. Discussion

The composition of intestinal microbes in mice with CLP-induced sepsis was altered by treatment with FX, and the OTU quantity of the intestinal flora in mice with CLP sepsis was altered by IRF3. FX changed the species diversity and species distribution of intestinal microbes in mice with sepsis, and the similarity in intestinal microbial communities among all groups was low. In the gut of mice with CLP sepsis, FX treatment significantly increased the abundance of beneficial bacteria, such as Verrucomicrobiota and *Akkermansia*. Simultaneously, following FX treatment, the dominant intestinal bacteria in mice with CLP sepsis shifted from *Akkermansia*, *Escherichia*, and *Morganella* to *Akkermansia* and *Escherichia*. According to PICRUSt2, FX affects the intestinal flora of mice with CLP-induced sepsis by altering glucose metabolism. Further analyses of the correlations between the intestinal flora, inflammatory factors, and SCFAs can reveal the marker florae that affect the inflammatory response. This analysis indicated that, following FX treatment, the intestinal flora of mice with sepsis exhibited interactions with inflammatory factors and SCFAs. Notably, the levels of acetic acid and propionic acid showed negative correlations with the expression levels of inflammatory factors IL-6, IL-1β, and TNF-α. Conversely, the abundances of *Morganella*, *Proteus*, *Escherichia*, and *Klebsiella* exhibited positive correlations with the expression levels of IL-6, IL-1β, and TNF-α that were induced by CLP and negative correlations with the production of acetic acid and propionic acid.

The intestinal lumen contains many intestinal microbiota that regulate intestinal immune homeostasis and affect the development and function of host immune cells [19]. The destruction of the intestinal microbiota integrity may increase susceptibility to sepsis [20]. The apoptosis of midgut epithelial cells has been observed in patients with sepsis and mouse models. This weakens the gut barrier, which in turn affects inflammation [21,22].

In the present study, the alteration of the intestinal microbial composition in mice with CLP sepsis was observed after FX treatment, and this change was linked to the expression of inflammatory factors. Consequently, we further evaluated the relationship between inflammatory factors and the intestinal microbiota. Following FX treatment, the dominant intestinal flora of mice with CLP-induced sepsis changed from *Akkermansia*, *Escherichia*, and *Morganella* to *Akkermansia* and *Escherichia*. Based on a Spearman correlation analysis, *Morganella* was positively correlated with IL-6, IL-1a, and TNFα expression after CLP and negatively correlated with the production of acetic and propionic acids.

The abundance of *Akkermansia*, a common anaerobic organism in the human and rodent intestinal microbiome, is negatively correlated with inflammation in inflammatory bowel disease [23]. FX increased the abundance of *Akkermansia*, suggesting that FX has a positive effect on the structure and function of the intestinal flora in mice with sepsis. *Morganella morganii* is a Gram-negative bacterium that causes various infections, including sepsis, leading to high mortality rates [24]. The results of this study suggested that FX alleviates the CLP-induced dysregulation of the intestinal microflora and inhibits the expression of IL-1β, IL-6, and TNF-β.

SCFAs are produced by fermenting indigestible polysaccharides (e.g., dietary fibers). Approximately 10% of SCFAs are excreted in the stool after they are produced in the gastrointestinal tract, while the remaining SCFAs are absorbed to provide energy to the host epithelium. Via the portal vein and circulation, SCFAs are transported to other organs and play a systemic role [25,26,27]. In humans, acetate, propionate, and butyrate make up more than 95% of intestinal SCFAs [28,29,30]. FX facilitated an increase in acetic and propionic acids, thus attenuating inflammation by regulating SCFA production. FX may have effects against inflammatory and bacterial processes. However, the sequential relationship between the anti-inflammatory and anti-bacterial effects of FX warrants further investigation, especially in the context of intestinal microflora transplants in mice. The relationship between IRF3 and intestinal microbes was not within the scope of this study. Subsequent studies should focus on the impact of FX on molecular signal transduction mechanisms.

## 4. Materials and Methods

### 4.1. Chemicals and Reagents

AbMole (Shanghai, China) provided FX with over 98.5% purity, as determined using high-performance liquid chromatography. Other reagents were of domestic analytical purity. For the animal experiments, FX powder was dissolved in DMSO to prepare a working solution with a concentration of 1.0 mg/mL. The FX suspension was then administered intraperitoneally (i.p.) to subjects at a daily dosage of 1.0 mg/kg. The Control group received an equivalent volume of the vehicle.

### 4.2. Animals

For the animal experiments, pathogen-free C57BL/6 mice (8–10 weeks old) were used with an equal distribution of males and females. Wild-type (WT) mice were obtained from Shanghai SLAC Laboratory Animal Co., Ltd. (Shanghai, China), and *Irf3^−/−^* mice were sourced from the RIKEN BioResource Research Center (BRC-No:00858, Tokyo, Japan). The mice had unrestricted access to food and water. Experimental conditions included a temperature range of 23–25 °C, humidity levels of 40–60%, and a 12 h light/dark cycle. Male mice weighing 20–22 g and female mice weighing 18–20 g were randomly assigned to groups. All animal experiments were conducted in accordance with the *Guide for the Care and Use of Laboratory Animals*. The study protocol was approved by the Institutional Animal Care and Use Committee of the Fujian Normal University (approval no. 201800013).

### 4.3. Model of CLP-Induced Sepsis

Sepsis was induced using a previously described CLP model [17]. In the Control group, laparotomies were performed without ligation or puncture.

### 4.4. Experimental Protocol

Eighty mice (female, n = 40; male, n = 40) were randomly allocated to four groups: Control, Control + FX, CLP-induced sepsis (CLP), and CLP + FX.

To effectively assess the efficacy of sepsis treatment, a mortality rate of 50% was required in mice with CLP-induced sepsis. Mice were administered FX (1.0 mg/kg/day) 2 h after the establishment of the sepsis model. Mice were injected with pentobarbital sodium salt intraperitoneally, and tissue, feces, and ascites samples were collected [18,30].

### 4.5. Quantitative Reverse Transcription PCR (RT-qPCR)

TRIzol reagent (Takara, Tokyo, Japan) was used to isolate total RNA from serum and lung tissues. RT-qPCR was conducted following previously described methods [16] and using primers listed in Table 1 for mRNA amplification.

### 4.6. Enzyme-Linked Immunosorbent Assay (ELISA)

Cytokines IL-1β, IL-6, and TNF-β were quantified using ELISA kits (IL-6: SM6000B; IL-1β: SMLB00C; TNF-β: SMTA00B; R&D Systems, Minneapolis, MN, USA) as per the manufacturer’s protocols.

### 4.7. Acquisition of Ascites in Mice

Two hours after CLP modeling, mice were subjected to the intraperitoneal injection of FX solutions at a dosage of 1.0 mg/kg/day or a control solvent lacking any active ingredient. After a 24 h treatment period, the mice were anesthetized, euthanized, and securely positioned on the operating table. Within a biosafety cabinet, 1 mL of sterile saline solution was injected into the abdominal cavities of the mice. Subsequently, the abdomen was gently massaged, and saline was extracted using a syringe, with caution not to puncture the organs or intestines. The collected saline was then transferred to a sterile EP tube for further processing.

### 4.8. Detection of Bacteria in Mouse Abdominal Cavity

Ascites samples were diluted at a specific ratio; subsequently, 100 µL was inoculated into the agar medium. The culture was incubated at 37 °C for 24 h, during which the growth of colonies was observed and quantified.

### 4.9. Quantification of Fecal Short-Chain Fatty Acids (SCFAs)

To quantify fecal SCFAs, 50 ± 1 mg of feces was added to 1 mL of deionized water. The mixture was homogenized for 4 min at 40 Hz and centrifuged at 16,000× *g* for 30 min at 4 °C. In total, 0.8 mL of the supernatant was mixed with 0.1 mL 50% H_2_SO_4_ and 0.8 mL of 2-methylvaleric acid (25 mg/L stock in methyl tertbutyl ether) as an internal standard and stored at −20 °C after ultrasonication and centrifugation at 12,000× *g* for 10 min. The supernatant was used for gas chromatography–mass spectrometry (GC-MS) analysis (Shimadzu, Japan) using an autosampler with an injection volume of 1.0 µL in accordance with a previously described method [27].

### 4.10. DNA Extraction and Barcoded Sequencing of the 16S rRNA Gene

Fecal samples were collected from each mouse at 24 h after FX treatment and sent to Biomarker Technologies Co., Ltd. (Beijing, China) for DNA extraction and 16S rRNA gene sequencing. Library construction, sequencing, and data analysis were performed using the PacBio Sequel II platform (Biomarker Technologies Co., Ltd.).

### 4.11. Bioinformatic Analysis

Bioinformatics analyses were performed using BMK Cloud (Biomarker Technologies Co., Ltd.). Sequences with ≥97% similarity were clustered into operational taxonomic units (OTUs) using USEARCH (v10 ≥ 0), and OTUs with an abundance of <0.005% were filtered [28]. Alpha and beta diversities were analyzed at the OTU level using QIIME. Alpha diversity was characterized using Shannon, Acer, Chao1, and Simpson metrics. The differences in microbial composition were further characterized by calculating beta diversity and analyzed based on weighted UniFrac distances. Group differences were compared using Adonis, and the results of the analysis of similarity (ANOSIM) were visualized using principal coordinate analysis (PCoA). For different species, the heat map reflects the similarities and differences in composition between multiple samples based on colors. Using a threshold linear discriminant analysis score of ≥4, the effect size of linear discriminant analyses can identify biomarkers with significant differences between groups and analyze the evolutionary relationships between species. In the plot, dots with different colors represent microbiomes with significant differences in the corresponding groups; ossia indicates significant differences between groups, while light yellow dots indicate a lack of significant influence. Finally, the sequences were compared with data from the Kyoto Encyclopedia of Genes and Genomes, and functional predictions were made using the phylogenetic investigation of communities via the reconstruction of unobserved state 2 (PICRUSt2).

### 4.12. Statistical Analysis

Data are expressed as the mean ± standard deviation. All results were analyzed using Tukey’s post hoc tests and one-way analysis of variance (ANOVA). Images were processed using Photoshop (Illustrator 2020; Adobe, San Jose, CA, USA) and ImageJ v1.8.0 (National Institutes of Health, Bethesda, MD, USA). GraphPad Prism (v8.0; GraphPad Software, San Diego, CA, USA) was used to perform statistical analyses. *p* < 0.05 was set as the significance level.

## 5. Conclusions

In this study, FX elevated the levels of acetic acid and propionic acid in the intestinal tract of mice with sepsis. Additionally, FX modulated the intricate interplay between the intestinal flora, inflammatory factors, and SCFAs, thereby contributing to the therapeutic effect of FX in mice with sepsis. Nevertheless, further exploration is warranted to discern the chronological sequence of the effects of FX on the intestinal flora, inflammatory factors, and SCFAs in the context of sepsis treatment.

## Figures and Tables

**Figure 1 ijms-24-13803-f001:**
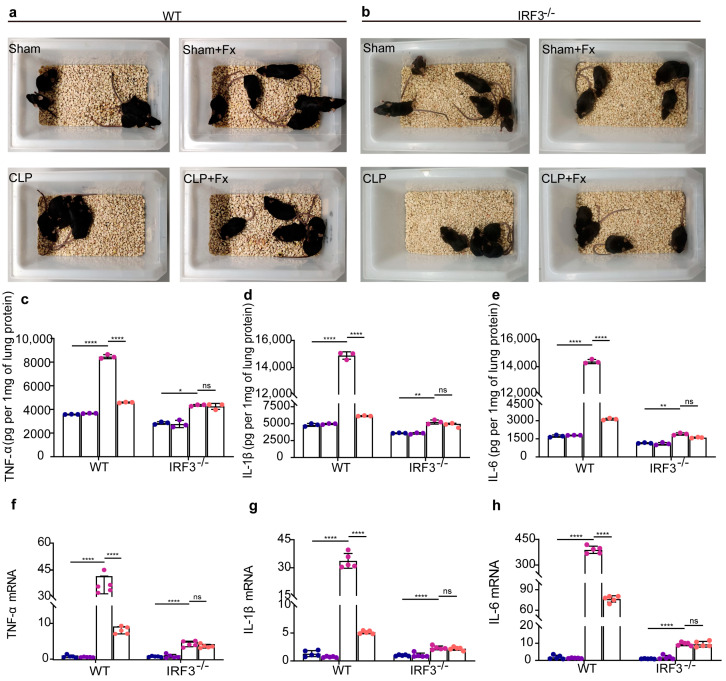
Fucoxanthin (FX) exhibited therapeutic effects in WT and *Irf3^−/−^* mice with cecal ligation and puncture (CLP)-induced sepsis. (**a**,**b**) Schematic diagram of the state of WT (**a**) and *Irf3^−/−^* (**b**) mice after 24 h of treatment. (**c**−**h**) FX inhibits inflammatory cytokines via IRF3 in a mouse model of CLP-induced sepsis; (**c**−**e**) concentrations of TNF-α (**c**), IL-1β (**d**), and IL-6 (**e**) in the lung tissues of WT and *Irf3*^−/−^ mice were determined using ELISA; (**f**–**h**) mRNA levels of *TNF-α* (**f**), *IL-1β* (**g**), and *IL-6* (**h**) in the lung tissues of WT and *Irf3^−/−^* mice were determined using RT−qPCR. The data were analyzed using ANOVA and Tukey’s post hoc tests; * *p* < 0.05, ** *p* < 0.01, and **** *p* < 0.0001. ns, not significant. Results are representative of at least three independent experiments. Blue represents Sham, purple represents Sham + FX, pink represents CLP, orange represents CLP + FX, and each point represents one mouse.

**Figure 2 ijms-24-13803-f002:**
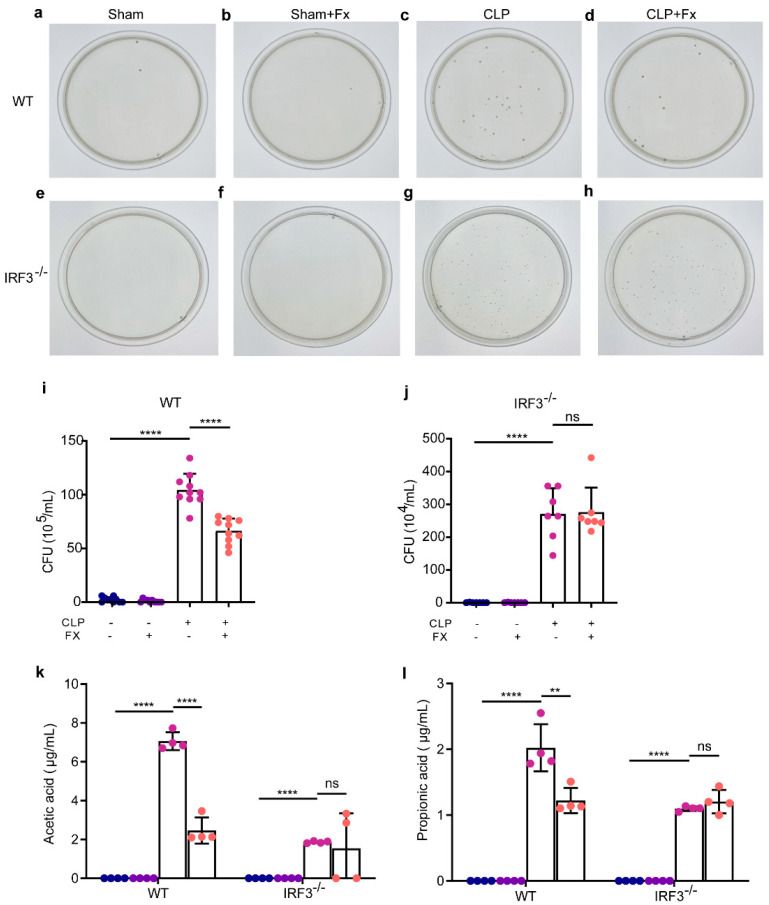
Effects of FX on the intraperitoneal bacterial load. (**a**–**d**) Growth of intraperitoneal bacterial colonies in WT mice; (**e**–**h**) growth of intraperitoneal bacterial colonies in *Irf3^−/−^* mice; (**i**) intraperitoneal load statistics in WT mice (n = 10); (**j**) intraperitoneal load statistics in *Irf3*^−/−^ mice (n = 7). (**k**) Effect of FX on short-chain fatty acid acetic acid in the ascites of mice with CLP sepsis (n = 4); (**l**) effect of FX on propionic acid in the ascites of mice with CLP sepsis (n = 4). The data were analyzed using ANOVA and Tukey’s post hoc test; ** *p* < 0.01, and **** *p* < 0.0001. ns, not significant. Results are representative of at least three independent experiments.

**Figure 3 ijms-24-13803-f003:**
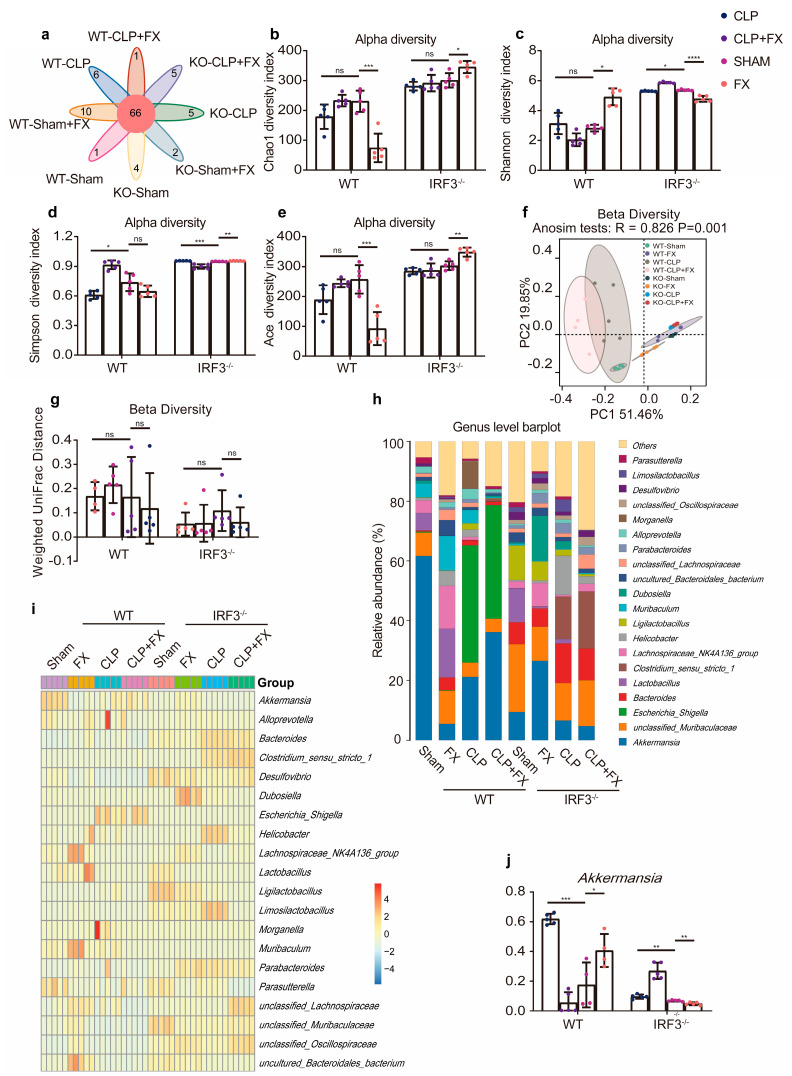
FX remodels the intestinal microbiota in mice with CLP-induced sepsis via IRF3. (**a**) Venn diagram showing the OTUs of intestinal microorganisms in WT and *Irf3*^−/−^ mice. (**b**–**e**) Alpha diversity of the intestinal microbiota at the OTU level. (**b**) Chao1, (**c**) Shannon (**d**) Simpson, and (**e**) Ace indices. (**f**,**g**) Beta diversity of the intestinal microbiota at the OTU level. (**f**) Beta diversity PCoA plots based on weighted UniFrac Adonis analysis in distinct groups. (**g**) Beta diversity based on weighted UniFrac ANOSIM in distinct groups. (**h**) Histogram showing the species distribution at the genus level. (**i**) Heat map analysis of the relative abundance of intestinal microorganisms in distinct groups at the genus level. (**j**) Relative abundance of *Akkermansia* spp. in distinct groups of WT and *Irf3*^−/−^ mice at the genus level. The heat map shows the Z-value obtained after the standardization of the relative abundance of species in each row. The data were analyzed using ANOVA and Tukey’s post hoc tests. Data are expressed as means ± SD. * *p* < 0.05, ** *p* < 0.01, *** *p* < 0.001, and **** *p* < 0.0001; ns, not significant. Results are representative of at least three independent experiments.

**Figure 4 ijms-24-13803-f004:**
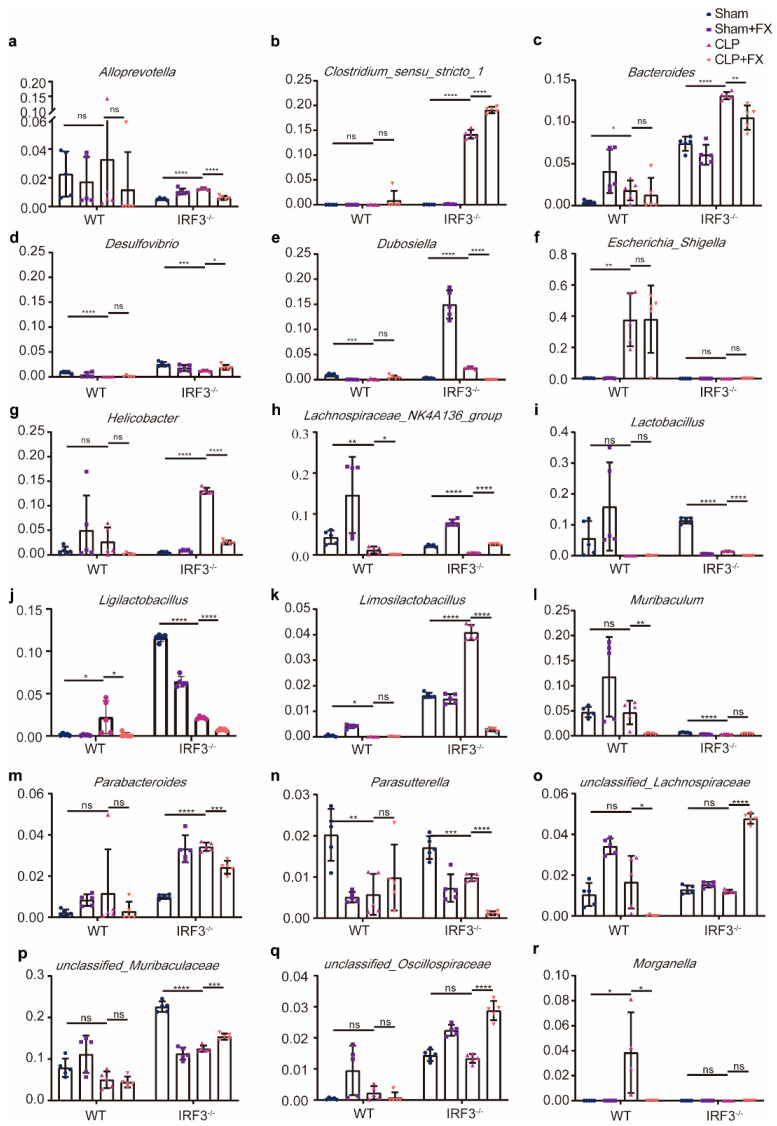
Relative abundances of intestinal microorganisms in different groups of WT and *Irf3*^−/−^ mice at the genus level. (**a**) Relative abundances of *Alloprevotella*; (**b**) *Clostridium* sensu stricto 1; (**c**) *Bacteroides*; (**d**) *Desulfovibrio*; (**e**) *Dubosiella*; (**f**) *Escherichia*; *Shigella*; (**g**) *Helicobacter*; (**h**) Lachnospiraceae NK4A136 group; (**i**) *Lactobacillus*; (**j**) *Ligilactobacillus*; (**k**) *Limosilactobacillus*; (**l**) *Muribaculum*; (**m**) *Parabacteroides*; (**n**) *Parasutterella*; (**o**) unclassified Lachnospiraceae; (**p**) unclassified Muribaculaceae; (**q**) unclassified Oscillospiraceae; (**r**) *Morganella*. The data were analyzed using ANOVA and Tukey’s post hoc tests; * *p* < 0.05, ** *p* < 0.01, *** *p* < 0.001, and **** *p* < 0.0001. ns, not significant. Results are representative of at least three independent experiments.

**Figure 5 ijms-24-13803-f005:**
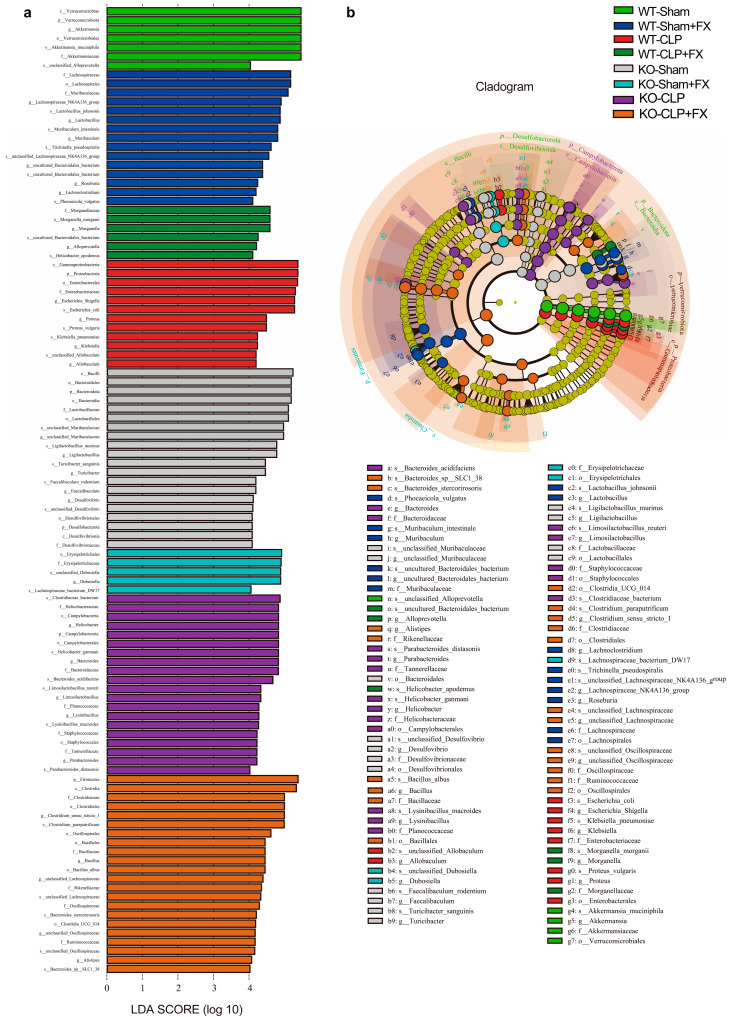
LEfSe analysis of intestinal microbiological differences among groups of WT and *Irf3*^−/−^ mice. (**a**) LEfSe statistics were used to compare celiac bacteria between groups (LDA > 4.0); (**b**) evolutionary branching diagram based on the LEfSe analysis.

**Figure 6 ijms-24-13803-f006:**
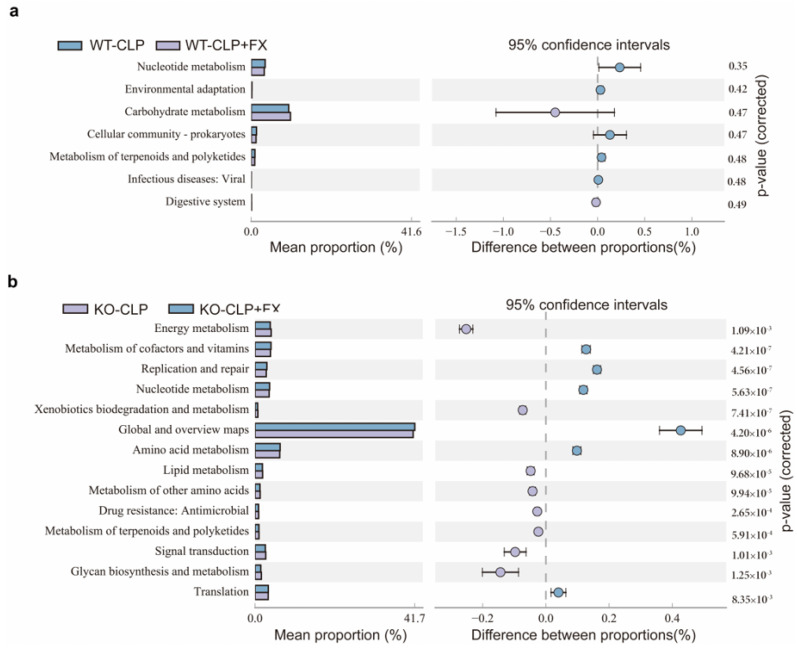
Functional prediction of intestinal microorganisms in WT and *Irf3*^−/−^ mice. (**a**) PICRUSt prediction of intestinal microbial functions in the CLP and CLP + FX groups of WT mice; (**b**) PICRUSt prediction of intestinal microbial functions in the CLP and CLP + FX groups of *Irf3^−/−^* mice.

**Figure 7 ijms-24-13803-f007:**
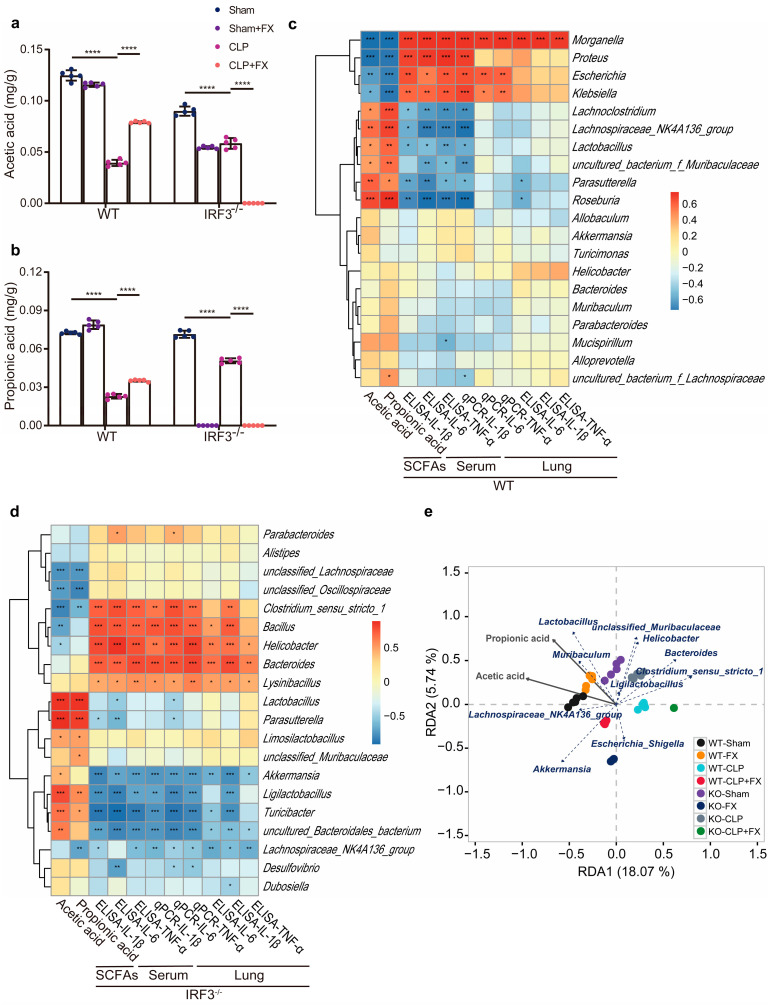
Intestinal flora and environmental factors in CLP-induced sepsis in WT and *Irf3*^−/−^ mice. (**a**,**b**) Effects of FX on acetic acid (**a**) and propionic acid (**b**) levels produced by intestinal microorganisms in WT and *Irf3^−/^*^−^ mice with CLP-induced sepsis (n = 5). (**c**) A Spearman’s rank correlation heat map for differences between the bacterial genera in the levels of SCFAs, IL-6, IL-1β, and TNF-α in WT mice with CLP-induced sepsis. (**d**) A Spearman’s rank correlation heat map showing differences between the bacterial genera in the levels of SCFAs, IL-6, IL-1, and TNF in *Irf3^−/−^* mice with CLP-induced sepsis. Spearman’s coefficients are represented in different colors. (**e**) Analysis of redundant bacterial genera and SCFAs in WT and *Irf3^−/−^* mice with CLP-induced sepsis. The data were analyzed using ANOVA and Tukey’s post hoc tests; * *p* < 0.05, ** *p* < 0.01, *** *p* < 0.001, and **** *p* < 0.0001. ns, not significant. Results are representative of at least three independent experiments.

**Table 1 ijms-24-13803-t001:** Sequences of primers used in RT-qPCR.

Primer	Sequence (5′–3′)
TNF-α	F: GCCTCCCTCTCATCAGTTCTA
TNF-α	R: GGCAGCCTTGTCCCTTGA
IL-6	F: CTTGGGACTGATGCTGGTG
IL-6	R: TCATTTCCACGATTTCCCA
IL-1β	F: TCATTGTGGCTGTGGAGAAG
IL-1β	R: TCATCTCGGAGCCTGTAGTG

IL: interleukin; TNF: tumour necrosis factor; F: forward, R: reverse.

## Data Availability

All data included in this study are available upon request by contacting the corresponding author.

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
