# Peer review of "Fucoxanthin Ameliorates Sepsis via Modulating Microbiota by Targeting IRF3 Activation"

_ijms, 2023, doi:10.3390/ijms241813803_

Round 1

Reviewer 1 Report

Su et al in the present manuscript demonstrated that Fucoxanthin (FX) suppresses pro-inflammatory cytokines and modulates the gut microbiome through IRF3. Authors further suggested that FX reduces the peritoneal bacterial load and increases the SCFAs like acetic and propionic acids in CLP mice. FX increases the beneficial bacterial species Verrucomicrobiota and Akkermansia while reduces harmful Morganella spp. and found positive correlation between Morganella spp and pro-inflammatory cytokines (IL-6, IL-1β, TNF-α) induced by CLP.

Specific comments:

The manuscript has throughout typological errors. Please read the manuscript carefully and correct it accordingly.

In Results section

Please abbreviate at first place e.g. line 74, cecal ligation and puncture (CLP) is abbreviated after using term CLP in line 68. Please abbreviate OTU.

Line 98 and 99, “the ascites of WT mice was diluted 105 times, whereas 98 that of Irf3-/- mice was diluted 104 times”. Please correct 105 and 104, and change it to 105 and 104.

Figure 1 Please mention the symbols for bar graph (C-H). Each color symbol denotes which group? I assume that it must be Sham, Sham+Fx, CLP and CLP+Fx. But still need to show clearly symbols for the bar graph. Figure 1C non-significant (ns) is missing in IRF3-/- between CLP vs CLP+Fx.

Figure legend 1 (C-E) “Serum concentrations of TNF-α (C), IL-1β (D), and IL-6 (E) in the lung tissues of WT and Irf3-/- mice”. Since, authors measured the cytokines in the lung tissue, then it is not serum concentrations and so, “serum” word should be deleted.

Figure 2 (A-H) photographs for WT and IRF3-/- mice showing the bacterial load is not very clear especially for IRF3-/- and it is hard to visualize the differences in colony numbers between different groups.

Figure legend 2: Line 118, “Effects of FX on the expression of intraperitoneal bacteria contents”. There is no expression studied here, it is just a count of bacteria. So, please delete word expression instead you can write as “Effects of FX on intraperitoneal bacteria load”.

Line 127, heading: 2.3 FX can effectively reduce the content of acetic acid and propionic acid in the abdominal water of CLP sepsis mice through IRF3. Instead of using “abdominal water” use term “peritoneal lavage”.

Line 138 FX was demonstrated to have a strong inhibitory effect on acetic acid and propionic acid content in the abdominal water of CLP sepsis “rats”, mediated by IRF3. I assume this typo error as it should be mice instead of rats.

Figure 3, FX treatment alone in the WT or IRF3-/- mice was given in naïve unmanipulated mice or sham mice? As, in the figure, it is presented as WT+FX. Because, the proper control is Sham+FX.

Line 179, According to Figure 3(H–J) and “4”, Akkermansia spp. dominated the sham group of both WT and Irf3-/- mice. There is no figure for Akkermansia spp. in Figure 4. So, Fig. 4 should be deleted from the text.

In Irf3-/- mice, CLP + FX mice did not display a notably higher abundance of Akkermansia spp. than “Irf3-/-mice” (P > 0·05). Is this comparison between the IRF3-/- vs WT or CLP + FX IRF3-/- mice vs. CLP IRF3-/- mice? If this is CLP + FX IRF3-/- mice vs. CLP IRF3-/- mice, then Figure 3J shows significance (p<0.01). Please clarify and correct it.

Line 194, The microbiota in the Sham, CLP, and Alo treatment groups. What is Alo group? Please clarify. Is it Alo means FX alone?

Figure 5 is too crowded as it is difficult to follow and read.

There are two results of acetic and propionic acid. 1.) Figure 2K-L in peritoneal lavage 2.) Figure 7A, B in fecal matter. But it is not clearly mentioned in the figure legends that where these SCFAs were measured. Please clearly mention the source in the figure legends.

In Discussion, Line 285, 455 Instead of “he” findings. “The” findings indicated.

There is a repetition of same lines in the discussion and conclusion. Try to concise the conclusion with main message.

In Methods, which grade of CLP was done? Is it mid or high ligation? Differences in the ligation results in mid-grade or severe-sepsis. Line 365, To effectively assess the efficacy of sepsis treatment, a mortality rate of 50% was required in mice with CLP-induced sepsis. I think it is mid-grade sepsis. Please mention it clearly.

The manuscript has typological errors throughout. English should be improved.

Scientific terms should be used e.g. peritoneal lavage instead of abdominal water and so on.

Author Response

August 26, 2023

Prof. Dr. Maurizio Battino

Editor-in-Chief

Ms. Chelsaea Xu

Assistant Editor

Dear Editor,

I wish to re-submit the manuscript titled “Fucoxanthin ameliorates sepsis through modulating microbiota by targeting IRF3 activation.” The manuscript ID is ijms-2577588.

We would like to extend our sincere appreciation to both you and the reviewers for the invaluable suggestions and insightful perspectives provided. The manuscript has greatly benefited from these astute recommendations.

Attached is the revised version of our manuscript. In the following pages are our point-by-point responses to each of the comments of the reviewers. Revisions in the text are highlighted by the utilization of the color red. We hope that the revisions in the manuscript and our accompanying responses would be sufficient to make our manuscript suitable for publication in International Journal of Molecular Sciences.

Thank you for your consideration. I look forward to hearing from you.

Sincerely,

Jingqian Su, Ph.D.

Associate Professor

Fujian Key Laboratory of Innate Immune Biology

Biomedical Research Center of South China

College of Life Science, Fujian Normal University

Fuzhou 350117, Fujian, China

Tel: +86-18950498937

E-mail: sjq027@fjnu.edu.cn

Responses to the comments of Reviewer #1

  1. The manuscript has throughout typological errors. Please read the manuscript carefully and correct it accordingly.

Response:

We would like to express our profound appreciation to the reviewer for the invaluable suggestions. The manuscript has been carefully read and corrected. The manuscript underwent editing by Editage, a professional language editing company.

  1. Please abbreviate at first place e.g. line 74, cecal ligation and puncture (CLP) is abbreviated after using term CLP in line 68. Please abbreviate OTU.

Response:

We express our sincere gratitude for dedicating your time to thoroughly review this manuscript. We greatly appreciate your valuable feedback. Considering the initial comment mentioned, we have made necessary revisions at the original location. Specifically, we have included a comprehensive explanation of the first instance of CLP in line 43, and subsequently employed abbreviations in the subsequent text. The abbreviation "OUT" has been introduced after its full name on line 148 of this manuscript and is consistently utilized in its abbreviated form thereafter.

  1. Line 98 and 99, “the ascites of WT mice was diluted 105 times, whereas 98 that of Irf3-/- mice was diluted 104 times”. Please correct 105 and 104, and change it to 105and 104.

Response:

Thank you for your comment and we apologize for this error. A superscript correction has been made to address the formatting issues mentioned above in Lines 90-91.

  1. Figure 1 Please mention the symbols for bar graph (C-H). Each color symbol denotes which group? I assume that it must be Sham, Sham+Fx, CLP and CLP+Fx. But still need to show clearly symbols for the bar graph. Figure 1C non-significant (ns) is missing in IRF3-/- between CLP vs CLP+Fx.

Response:

We wish to extend our sincere gratitude to the reviewers for their invaluable suggestions. The following prompt has been added to the caption of Figure 1: blue represents Sham, purple represents Sham+FX, pink represents CLP, orange represents CLP+FX, the number of points represents the number of mice, and each point represents one mouse. The non-significant (ns) has added in Figure 1c.

  1. Figure legend 1 (C-E) “Serum concentrations of TNF-α (C), IL-1β (D), and IL-6 (E) in the lung tissues of WT and Irf3-/- mice”. Since, authors measured the cytokines in the lung tissue, then it is not serum concentrations and so, “serum” word should be deleted.

Response:

We would like to express our profound appreciation to the editors for their invaluable suggestions. The term 'serum' has been deleted from the annotations in Figure 1 (C-E).

  1. Figure 2 (A-H) photographs for WT and IRF3-/- mice showing the bacterial load is not very clear especially for IRF3-/- and it is hard to visualize the differences in colony numbers between different groups.

Response:

We express our sincere gratitude for your valuable correction, as it holds significant importance for our research. Following your guidance, we have thoroughly reevaluated Figure 2 and successfully addressed the concern you raised. Consequently, we have updated the figure, resulting in a more lucid representation of the bacterial carrier map. Furthermore, we have taken the initiative to provide high-definition images of the bacterial carriers within the system, facilitating enhanced visual examination.

  1. Figure legend 2: Line 118, “Effects of FX on the expression of intraperitoneal bacteria contents”. There is no expression studied here, it is just a count of bacteria. So, please delete word expression instead you can write as “Effects of FX on intraperitoneal bacteria load”.

Response:

We wish to extend our sincere gratitude to the reviewers for their invaluable suggestions. In line 107, the caption of Figure 2 has been modified to read “Effects of FX on intraepithelial bacteria load”.

  1. Line 127, heading: 2.3 FX can effectively reduce the content of acetic acid and propionic acid in the abdominal water of CLP sepsis mice through IRF3. Instead of using “abdominal water” use term “peritoneal lavage”.

Response:

We express our sincere gratitude for your valuable correction. The subheading has been changed from 'domestic water' to 'permanent lavage' in line 114.

  1. Line 138 FX was demonstrated to have a strong inhibitory effect on acetic acid and propionic acid content in the abdominal water of CLP sepsis “rats”, mediated by IRF3. I assume this typo error as it should be mice instead of rats.

Response:

Thank you for your comment and we apologize for this error. In Line 124, we have corrected rates to mice.

  1. Figure 3, FX treatment alone in the WT or IRF3-/- mice was given in naïve unmanipulated mice or sham mice? As, in the figure, it is presented as WT+FX. Because, the proper control is Sham+FX.

Response:

We would like to extend our heartfelt appreciation for the invaluable suggestion offered by the reviewer. FX treatment was performed during sham surgery. In order to express more accurately, the WT+FX has been changed to WT-Sham+FX in Figure 3a.

  1. Line 179, According to Figure 3(H–J) and “4”, Akkermansia spp. dominated the sham group of both WT and Irf3-/- mice. There is no figure for Akkermansia spp. in Figure 4. So, Fig. 4 should be deleted from the text.

Response:

Thank you for your comment and we apologize for this error. In Line 158, Fig. 4 has been deleted from the text.

  1. In Irf3-/- mice, CLP + FX mice did not display a notably higher abundance of Akkermansia spp. than “Irf3-/-mice” (P > 0·05). Is this comparison between the IRF3-/- vs WT or CLP + FX IRF3-/- mice vs. CLP IRF3-/- mice? If this is CLP + FX IRF3-/- mice vs. CLP IRF3-/- mice, then Figure 3J shows significance (p<0.01). Please clarify and correct it.

Response:

We express our gratitude for your inquiry. We have duly examined the raw data pertaining to the plot presented in Figure 3j and conducted a thorough analysis. The process of data verification has revealed a noteworthy disparity between the CLP and CLP+FX groups in Irf3-/- knockout mice, with the abundance of Akkermansia in CLP+FX group exhibiting a significantly lower level compared to that observed in CLP group.

  1. Line 194, The microbiota in the Sham, CLP, and Alo treatment groups. What is Alo group? Please clarify. Is it Alo means FX alone?

Response:

Thank you for your comment and we apologize for this error. We have corrected Alo to FX in lines 172-183.

  1. Figure 5 is too crowded as it is difficult to follow and read

Response:

We express our sincere gratitude for your valuable suggestion, as it holds significant importance to us. Considering the elongated structure of Figure 5, dividing it would not enhance clarity. After thorough deliberation, we have reached the decision to refrain from splitting it. Nevertheless, we will offer high-resolution supplementary images for the convenience of editors, reviewers, and readers, enabling them to examine the details more closely.

  1. There are two results of acetic and propionic acid. 1.) Figure 2K-L in peritoneal lavage 2.) Figure 7A, B in fecal matter. But it is not clearly mentioned in the figure legends that where these SCFAs were measured. Please clearly mention the source in the figure legends.

Response:

We express our sincere gratitude for your valuable suggestion. In this manuscript, we have included additional information regarding the measurement location and source of short-chain fatty acids (SCFA). To prevent any ambiguity with SCFA originating from intestinal organisms, we have added annotations to Figures 2K and L (Lines 109-110), elucidating their origin in ascites.

  1. In Discussion, Line 285, 455 Instead of “he” findings. “The” findings indicated.

There is a repetition of same lines in the discussion and conclusion. Try to concise the conclusion with main message.

Response:

We express our sincere gratitude for the invaluable feedback provided. In response, we have implemented necessary amendments at line 261 of the manuscript to enhance its accuracy. Furthermore, we have diligently revised and succinctly condensed the discourse and conclusion section, employing concise language to effectively encapsulate our findings within lines 407-413.

  1. In Methods, which grade of CLP was done? Is it mid or high ligation? Differences in the ligation results in mid-grade or severe-sepsis. Line 365, To effectively assess the efficacy of sepsis treatment, a mortality rate of 50% was required in mice with CLP-induced sepsis. I think it is mid-grade sepsis. Please mention it clearly.

Response:

We appreciate your inquiry. It is crucial to acknowledge the significance of various ligation positions and needle models, as they can exert a substantial influence on the cecal ligation and puncture (CLP) model. In this study, a 22G needle was employed for puncturing and ligating the cecum at a distance of one-third from the tip. This procedure induced a state of moderate sepsis.

Reviewer 2 Report

The current fundamental study is aimed to elucidate how FX-targeted IRF3 modulation of intestinal microbiota composition, influencing sepsis development.

The conclusion section should be revised - it significantly copies the discussion section and must be more concise.

The reference list must be updated and some valuable recent investigations can be added:

https://www.reanimatology.com/rmt/article/view/1833

https://www.reanimatology.com/rmt/article/view/2134

https://www.reanimatology.com/rmt/article/view/1917

Author Response

August 26, 2023

Prof. Dr. Maurizio Battino

Editor-in-Chief

Ms. Chelsaea Xu

Assistant Editor

Dear Editor,

I wish to re-submit the manuscript titled “Fucoxanthin ameliorates sepsis through modulating microbiota by targeting IRF3 activation.” The manuscript ID is ijms-2577588.

We would like to extend our sincere appreciation to both you and the reviewers for the invaluable suggestions and insightful perspectives provided. The manuscript has greatly benefited from these astute recommendations.

Attached is the revised version of our manuscript. In the following pages are our point-by-point responses to each of the comments of the reviewers. Revisions in the text are highlighted by the utilization of the color red. We hope that the revisions in the manuscript and our accompanying responses would be sufficient to make our manuscript suitable for publication in International Journal of Molecular Sciences.

Thank you for your consideration. I look forward to hearing from you.

Sincerely,

Jingqian Su, Ph.D.

Associate Professor

Fujian Key Laboratory of Innate Immune Biology

Biomedical Research Center of South China

College of Life Science, Fujian Normal University

Fuzhou 350117, Fujian, China

Tel: +86-18950498937

E-mail: sjq027@fjnu.edu.cn

Responses to the comments of Reviewer #2

  1. The conclusion section should be revised - it significantly copies the discussion section and must be more concise.

Response:

We express our sincere gratitude for your kind effort in sparing your valuable time to offer constructive feedback on this manuscript, which holds significant importance for the article. In response, we have made revisions and provided concise summaries in the conclusion section (Lines 407-413).

  1. The reference list must be updated and some valuable recent investigations can be added: https://www.reanimatology.com/rmt/article/view/1833

https://www.reanimatology.com/rmt/article/view/2134

https://www.reanimatology.com/rmt/article/view/1917

Response:

Thank you for your valuable feedback and information that has enriched our article. In this manuscript, the references have been added in the background sections (Reference 3, 4 and 11).
